# Anti-Obesity Effect of Polygalin C Isolated from *Polygala japonica* Houtt. via Suppression of the Adipogenic and Lipogenic Factors in 3T3-L1 Adipocytes

**DOI:** 10.3390/ijms221910405

**Published:** 2021-09-27

**Authors:** Wona Jee, Seung-Hyeon Lee, Hyun Min Ko, Ji Hoon Jung, Won-Seok Chung, Hyeung-Jin Jang

**Affiliations:** 1College of Korean Medicine, Kyung Hee University, 26 Kyungheedae-ro, Dongdaemun-gu, Seoul 02447, Korea; jee1ah@khu.ac.kr (W.J.); skyking27@khu.ac.kr (S.-H.L.); kocic77@khu.ac.kr (H.M.K.); johnsperfume@gmail.com (J.H.J.); omdluke@khu.ac.kr (W.-S.C.); 2Department of Science in Korean Medicine, Graduate School, Kyung Hee University, Seoul 02447, Korea

**Keywords:** Polygalin C, *Polygala japonica* Houtt., obesity, 3T3-L1, adipogenesis

## Abstract

Obesity is a risk factor for metabolic diseases including type 2 diabetes, nonalcoholic steatohepatitis (NASH), heart diseases, and cancer. This study aimed to investigate the anti-obesity effect of Polygalin C (PC) isolated from *Polygala japonica* Houtt. in 3T3-L1 adipocytes. Based on Oil Red O assay results, PC significantly decreased lipid accumulation compared to the control. We found that PC suppressed adipogenesis transcription factors including peroxisome proliferator-activated receptor γ (PPAR γ) and CCAAT/enhancer-binding protein (C/EBP) α, and lipogenic factors such as sterol regulatory element-binding protein 1c (SREBP 1c) and fatty acid synthase (FAS), in 3T3-L1 adipocytes using Western blotting and real-time polymerase chain reaction (PCR). Moreover, PC inhibited the differentiation of 3T3-L1 cells by regulating the AMP-activated protein kinase (AMPK)/acetyl-CoA carboxylase (ACC) and mitogen-activated protein kinase/protein kinase B (MAPK/Akt) signaling pathways. Additionally, we confirmed that PC inhibited early adipogenesis factors C/EBP β and C/EBP δ. Therefore, PC inhibited adipogenesis and lipogenesis in vitro. Thus, PC appears to exert potential therapeutic effects on obesity by suppressing lipid metabolism.

## 1. Introduction

Obesity, which is characterized by an increase in the number and size of adipocytes due to the imbalance between food intake and energy expenditure, is a major cause of chronic diseases, such as type 2 diabetes, high blood pressure, arteriosclerosis, arthritis, hyperlipidemia, simple hepatic steatosis, and non-alcoholic steatohepatitis (NASH) [1,2].

Anti-obesity drugs are important not only for the purpose of treatment for weight loss but also for the improvement of the aforementioned diseases. Several U.S. Food and Drug Administration (FDA)-approved synthetic drugs, as well as obesity therapy, have been reported to be helpful in weight loss and the management of metabolic diseases, but they have side effects when used for long periods [3,4,5]. Therefore, many researchers have studied the anti-obesity effects of natural products with superior safety [6,7]. In the development of obesity, differentiation from preadipocytes to adipocytes is an important process that causes hypertrophy and proliferation of adipose tissue [8,9]. Thus, inhibiting differentiation into adipocyte s may be a potential strategy for the treatment of obesity [10].

In general, 3T3-L1 adipocytes derived from mouse embryos are mainly used to investigate the mechanism of action related to differentiation [11]. Differentiation of adipocytes requires inducers, including 3-isobutyl-1-methylxanthine (IBMX), dexamethasone, and insulin (MDI medium) [12]. This cocktail synergistically regulates various adipogenic transcription factors, thereby inducing adipocyte differentiation [13]. CCAAT/enhancer-binding proteins (C/EBPs), of which there are six types, are also involved in the process of adipogenesis. Among all C/EBPs, C/EBP α, β, γ, and δ are associated with adipogenesis [14]. During differentiation induced by MDI, C/EBPβ and C/EBPδ activate the beginning of the adipogenic process, thereby inducing differentiation of adipocytes by upregulating the expression of peroxisome proliferator-activated receptor γ (PPARγ) and C/EBPα [15,16]. Acetyl-CoA carboxylase (ACC) and fatty acid synthase (FAS) are involved in lipogenesis, including fatty acid synthesis and lipid accumulation [17]. These factors determine the adipocyte phenotype [18]. AMP-activated protein kinase (AMPK) is a major factor in regulating energy balance in cells [19], but this protein also plays a role in adipogenesis by regulating sterol regulatory element-binding protein 1c (SREBP1c), which is involved in adipogenic and lipogenic processes [20,21]. The protein kinase B (Akt) and mitogen-activated protein kinase (MAPK) signaling pathways have overall effects on the differentiation process, leading to the expression of adipogenic factors [22,23,24].

The *Polygala* genus belongs to the Polygalaceae family. According to existing studies, *Polygala* exerts various pharmacological effects, such as trypanocidal activity [25] as well as anti-inflammatory [26], antinociceptive [27], antifungal [28], and anti-obesity [29] properties. *Polygala japonica* Houtt., which belongs to the *Polygala* genus, is widely distributed in Korea and Asia [30,31]. *P. japonica* Houtt., a perennial herbaceous plant widely used in traditional medicine that contains saponin, flavonoids, and xanthones as its main compounds, has anti-inflammatory, antibacterial, antidepressant, and sedative effects [32]. However, no studies have reported on the anti-obesity effect of Polygalin C (PC), a flavonoid isolated from *P. japonica* Houtt. Thus, we investigated whether PC has an anti-adipogenic effect in 3T3-L1 adipocytes in this study. We also evaluated the effect of PC on the regulation of the expression of key proteins involved in adipogenic and lipogenic pathways.

## 2. Results

### 2.1. Cytotoxicity of PC in 3T3-L1 Cells

We evaluated the effect of PC on the viability of adipocytes in 3T3-L1 cells using two types of cell viability assays, MTT assay and LDH release assay. As shown in Figure 1B,C, it was confirmed that cell viability was significantly decreased in adipocytes from the concentration of 75 μM. Based on these results, we concluded that PC treatment below 50 μM did not affect the viability of 3T3-L1 adipocytes and determined the optimal PC concentration for subsequent experiments.

### 2.2. Effects of PC on Adipocyte Differentiation in 3T3-L1 Preadipocytes

To investigate whether PC has an inhibitory effect on adipogenic differentiation, we used Oil Red O staining after cells were differentiated with or without various concentrations (0, 25, and 50 μM) of PC (Figure 1D). PC dose-dependently inhibited the production of lipid droplets in 3T3-L1 cells compared with fully differentiated adipocytes (Figure 1E,F). Thus, PC reduced the accumulation of lipid in 3T3-L1 cells during adipogenic differentiation.

### 2.3. Effect of PC on Adipogenic Transcription Factor Expression Levels

To examine how PC inhibited adipogenesis in 3T3-L1 cells, we first investigated the protein expression levels of key adipogenic markers. C/EBPα and PPARγ are well-known key transcription factors as regulators of the adipogenesis process. Therefore, we examined the expression of these factors by Western blotting. As shown in Figure 2A, the relative protein levels of both C/EBPα and PPARγ were significantly increased in MDI-induced adipocytes, but those of the PC groups decreased in a dose-dependent manner compared to the MDI-induced group. In addition, the protein expressions of the early adipogenesis factors, C/EBPβ and C/EBPδ, showed statistically significant differences in the PC-treated group at 50 μM compared with fully differentiated adipocytes (Figure 2B). Furthermore, we confirmed the expression levels of each adipogenic factor using real-time PCR (Figure 2C–F). As a result, it was confirmed that treatment with PC remarkably decreased the mRNA level of PPARγ, C/EBPα, C/EBPβ, and C/EBPδ, which were increased by MDI. Therefore, these results suggest that PC regulates adipogenesis by inhibiting C/EBPβ and C/EBPδ as well as suppressing the expression levels of the adipogenic transcription factors C/EBPα and PPARγ.

### 2.4. Effect of PC on Lipogenic Factor Expression Levels

To confirm whether PC inhibits lipogenic factors, we examined the expression levels of lipogenic transcription factors and enzymes by Western blotting and real-time PCR. FAS and SREBP1c play critical roles in lipogenesis, so we investigated their protein expression levels. PC significantly decreased the expression levels of FAS and SREBP1c in a dose-dependent manner compared to fully differentiated adipocytes (Figure 3A). mRNA levels also declined compared to those in fully differentiated adipocytes (Figure 3B,C). Furthermore, we observed a remarkable alleviating effect of PC on the reduction of MDI-induced SREBP1c expression in adipocytes via immunofluorescence assay (Figure 3D). These results indicate that PC regulates lipogenesis by inhibiting FAS and SREBP1c activation in fully differentiated adipocytes.

### 2.5. Effect of PC on AMPK/ACC and MAPK/Akt Signaling Pathway Expression

The MAPK/Akt and AMPK/ACC signaling pathways are involved in the general stages of adipogenesis. First, we investigated whether the activation of AMPK and ACC by PC plays an important role in the inhibition of adipocyte differentiation by Western blotting and real-time PCR. The total AMPK and ACC expression levels were not changed, and PC treatment significantly increased the phosphorylation of AMPK and ACC compared to fully differentiated adipocytes (Figure 4A). Additionally, we investigated PC modulation upon inhibition of AMPK activity. We used the AMPK inhibitor, compound C (compound C, 10 μM). Compound C treatment decreased the protein expression levels of *p*-AMPK and *p*-ACC, which was partially recovered in the PC co-treated group (Figure 4B).

Next, we investigated MAPK and Akt signaling in adipocytes with PC by Western blotting. MDI-induced adipocytes expressed high levels of activated MAPK kinases and Akt, among which Erk and Akt showed a particularly significant difference. Activated MAPK kinase and Akt were markedly decreased by PC, except for p38, in a dose-dependent manner (Figure 4B). Therefore, these results suggest that PC inhibits the general stages of adipocyte differentiation via the AMPK/ACC and MAPK/Akt signaling pathways.

## 3. Discussion

Obesity is defined as abnormal or excessive fat accumulation that poses a health risk [33]. According to the obesity and overweight fact sheet published by the World Health Organization in April 2020, the prevalence of obesity worldwide has nearly tripled since 1975. In 2016, more than 1.9 billion (39%) adults aged 18 or older were overweight, and more than 650 million (13%) were obese [34]. Obesity is a major risk factor for cardiovascular diseases (mainly heart disease, high blood pressure, and stroke), type 2 diabetes, cirrhosis (non-alcoholic steatohepatitis and non-alcoholic fatty liver disease), and cancer [35]. Recent studies have also shown that patients with chronic diseases associated with obesity are more vulnerable to COVID-19 [36,37].

According to the National Institute for Health and Care Excellence (NICE), medications are recommended in addition to a low-calorie diet and optimal physical exercise to maintain weight loss [38]. The most commonly used drugs are orlistat and liraglutide. Orlistat irreversibly inhibits pancreatic lipase, which breaks down dietary fat into absorbable free fatty acids, allowing fat to be excreted in the feces [39] and liraglutide acts as a GLP-1 receptor agonist and suppresses appetite by regulating insulin secretion [40]. However, these drugs commonly have side effects such as gastrointestinal disturbances, oily stools, and incontinence, and cases of acute pancreatitis have also been reported [41,42]. Therefore, long-term treatment of obesity with chemotherapy can have side effects, so many researchers are now discovering and evaluating effective and safe natural weight loss compounds [6,7]. Recently, bioactive compounds have been extensively studied, and provide not only essential roles as nutrients but also biochemical and pharmacological functions beneficial to human health [43]. In previous studies, Polygala genus plants have been reported to possess anti-obesity effects [29,44]. However, there are no studies on the anti-obesity activities and mechanisms of PC isolated from *P. japonica* Houtt. in adipocytes. Therefore, in this study, to evaluate the possibility of preventing and treating obesity, the anti-obesity effect of PC was investigated through experiments on MDI-induced 3T3-L1 adipocytes. To induce differentiation in 3T3-L1 cells, many studies have used a combination of IBMX (cAMP agonist), dexamethasone (glucocorticoid), and insulin [12].

3T3-L1 adipocytes derived from mouse embryos are commonly used to study metabolic disease conditions in the body. Unlike muscle and nerve cell differentiation, adipocyte differentiation is complicated by the interaction of various hormones and various transcription factors. The process of differentiating preadipocytes from adipocytes is accompanied by changes in cell morphology and gene expression patterns [11]. Most of these changes occur in adipocyte-specific transcription factors, depending on the differentiation process. Their primary role is the regulation of adipocyte differentiation, which is performed by C/EBPα, PPARγ, and SREBP1c. During adipocyte differentiation, these transcription factors are induced and interact with each other to regulate the expression of various adipocyte-specific genes, activate lipid metabolism, and induce adipocyte differentiation [45]. The C/EBP family comprises transcription factors with a basic leucine zipper domain, and there are six types of C/EBPs [46]. The expression of C/EBPβ and C/EBPδ is induced by MDI and cAMP, and their expression levels increase in the early stages of differentiation [14]. As a result, the expression levels of the adipogenic markers, C/EBPα and PPARγ, also increased then. Many studies have suggested the suppression of adipocyte differentiation via the downregulation of C/EBPα and PPARγ [47,48].

In order to confirm the anti-adipogenic effect of PC in MDI-induced 3T3-L1 cells, we examined the protein and mRNA expression levels of C/EBPα and PPARγ. In this study, PC treatment dose-dependently inhibited the expression levels of C/EBPα and PPARγ compared to those in the MDI-treated group. Additionally, early adipogenic factors C/EBPβ and C/EBPδ were decreased in a concentration-dependent manner compared to adipocytes at the mRNA level. Thus, PC appears to regulate adipogenesis by inhibiting early adipogenic factors, thereby suppressing the expression levels of adipogenic transcriptional factors C/EBPα and PPARγ.

FAS is regulated by SREBP1c and is a major lipogenic factor that catalyzes the biosynthesis of fatty acids [48]. Therefore, controlling the expression of FAS is important for obesity. PC decreased the expression of adipogenic enzymes; therefore, this study indicated that PC suppresses lipogenesis by inhibiting FAS.

The AMPK and MAPK/Akt pathways play important roles in the adipogenesis process. AMPK is the main factor that regulates the energy balance and cell cycle in cells [19]. However, this factor is also involved in adipogenesis and lipogenesis [20,21]. Activated AMPK, which is induced by increasing the intracellular AMP/ATP ratio, uptakes fatty acids and regulates downstream factors, such as triglycerides, cholesterol, and proteins related to metabolic enzymes and transcription factors in lipid metabolism due to β-oxidation. Among AMPK subtypes, AMPKα is mainly expressed in adipose tissue and controls lipid homeostasis. Increased expression of AMPKα prevents the precursor of SREBP 1c, which is a lipogenic factor, from breaking up [49]. In this study, the results confirmed that PC significantly upregulates the protein expression levels of *p*-AMPK/AMPK and *p*-ACC/ACC but downregulated the protein expression level of SREBP1c. The mRNA level also showed similar results to those of the protein expression level. In addition, we confirmed that PC suppresses the protein levels of ERK, JNK, and Akt phosphorylation except for p38. Bost et al. reported that p38 phosphorylation is involved in C/EBPα and PPARγ, and ERK1/2 and JNK are associated with adipocyte differentiation. Aside from some contradictory results that MAPK proteins are downregulated at some stage of the process from stem cells to adipocytes, ERK, p38, and JNK are upregulated during differentiation [50]. Thus, this study suggests that PC downregulates adipocyte differentiation through the AMPK and MAPK/Akt signaling pathways (Figure 5).

In this study, we demonstrated the anti-obesity effects of PC in MDI-induced 3T3-L1 cells. PC inhibited lipid accumulation as well as PPARγ, C/EBPα, SREBP1c, and FAS protein expression levels by upregulating the AMPK/ACC pathway and downregulating the MAPK/Akt signaling pathway in adipocytes. In addition, PC suppressed the mRNA levels of C/EBP δ and C/EBP β, which are early adipogenic factors. These findings provide an experimental basis for the anti-obesity effect of PC. Therefore, for the first time, we suggested the possibility of PC as a natural anti-obesity agent.

However, polyphenols such as polygalin C have low solubility and stability to heat, oxygen, and light, and can be easily degraded by the metabolism in the human body [51,52,53]. Therefore, it is necessary to analyze the stability of the Polygalin C structure against changes in temperature, time, and pH in vitro. In addition, animal research is required as further studies to analyze the stability of metabolites in the body. Finally, planning in vivo experiments, it is necessary to determine the dosage by considering several factors, such as the method of administration and body metabolism.

## 4. Materials and Methods

### 4.1. Chemicals

PC was obtained from Chemfaces (Wuhan Chemfaces Biochemical Co., Ltd., Wuhan, China). The structure is shown in Figure 1A.

### 4.2. Reagents

Dulbecco’s modified Eagle’s medium (DMEM) and fetal bovine serum (FBS) were purchased from Gibco (Grand Island, NY, USA). Phosphate-buffered saline (PBS) was purchased from Corning (Corning, NY, USA). IBMX, dexamethasone, insulin, and Oil Red O powder were obtained from Sigma-Aldrich (St. Louis, MO, USA). 3-(4,5-Dimethylthiazol-2-yl)-2,5-diphenyltetrazolium bromide (MTT) was purchased from Invitrogen (Carlsbad, CA, USA). Enhanced chemiluminescence solution (WestGlow Pico PLUS solA and solB) was purchased from Biomax Co., Ltd. (Seoul, Korea). Bradford protein assay reagent was purchased from Bio-Rad (Hercules, CA, USA). The EZ-LDH cytotoxicity assay kit was purchased from DoGenBio (ITSBIO, Seoul, Korea). Compound C was purchased from Sejin CI Co., Ltd. PPARγ, FAS, phospho-acetyl-CoA carboxylase (*p*-ACC), total ACC (*t*-ACC), phospho-AMPK (*p*-AMPK), total AMPK (*t*-AMPK), phospho-extracellular signal-regulated kinase (*p*-ERK), total ERK (*t*-ERK), phospho-c-Jun N-terminal kinase (*p*-JNK), total JNK (*t*-JNK), phospho-Akt (*p*-Akt), total Akt (*t*-Akt), phospho-p38 (*p*-p38), and total p38 (*t*-p38) antibodies were purchased from Cell Signaling Technology (Danvers, MA, USA). C/EBPα, C/EBPβ, C/EBPδ, SREBP1c, β-actin, anti-rabbit, and anti-mouse secondary antibodies were purchased from Santa Cruz Biotechnology (Dallas, TX, USA).

### 4.3. Cell Culture and Experimental Conditions

3T3-L1 cells and a mouse embryo fibroblast cell line were obtained from the American Type Culture Collection (Rockville, MD, USA). The cells were cultured in DMEM supplemented with 10% FBS and 100 U/mL penicillin in a cell incubator at 37 °C with 5% CO_2_. The cells were seeded at a density of 8 × 10^4^ cells/well in 6-well plates and replaced with media until confluence. For differentiation, the cells were incubated in differentiation medium (0.5 mM IBMX, 0.5 µM dexamethasone, and 1 µg/mL insulin). Then, the media was replaced with a maintenance medium containing 1 µg/mL insulin and PC (0–50 µM) once every 2 days. A schematic of the experimental process is shown in Figure 1C.

To assess the activity of AMPK about PC in the 3T3-L1 adipocytes, 3T3-L1 cells were pre-treated with 15 µM compound C for 1 hr, and then treated with or without PC. After that, the medium was replaced once every 2 days until Day 8.

### 4.4. Cell Viability

The cytotoxicity of Polygalin C in 3T3-L1 adipocytes was measured using an MTT solution or LDH cytotoxicity assay kit. 3T3-L1 preadipocyte cells were seeded in 96-well cell culture plates at 3 × 10^3^ cells/well and incubated until confluence. Then, the medium was replaced with a differentiation medium containing different concentrations of PC (0, 5, 10, 25, 50, 75, 100 μM). The medium was replaced with a medium containing 10 μg/mL insulin and different concentrations of PC for 8 days. After treatment, the cell viability of adipocytes was measured using a microplate reader (Bio-Rad).

### 4.5. Immunoblotting

Cells were lysed using cell lysis buffer (Cell Signaling Technology). Bio-Rad Protein Assay Reagent was used to measure protein concentrations for the Bradford assay. Each sample was separated by sodium dodecyl sulfate polyacrylamide gel electrophoresis using 8% acrylamide gel and then electro-transferred to a nitrocellulose membrane. The membranes were blocked with 3% bovine serum albumin (BSA) in tris-buffered saline/Tween-20 (TBS-T) at 4 °C for 1 h and incubated at 4 °C overnight with the following antibodies: C/EBP α, PPAR γ, SREBP 1c, FAS, *p*-ACC, *t*-ACC, *p*-AMPK, *t*-AMPK (1:2000), and β-actin (1:10,000). The membranes were extensively washed with TBS-T at least three times for 30 min and incubated with the following secondary antibodies for 1 h: goat anti-rabbit IgG-horseradish peroxidase (HRP) and goat anti-mouse IgG-HRP (all at 1:10,000 in TBS-T).

### 4.6. Oil Red O Assay

Treated cells were washed with PBS and fixed with 10% formalin for 1 h. Then, the cells were washed with 60% isopropanol and dried completely. A 60% Oil Red O working solution in distilled water (DW) was added to the cells. The stained cells were washed three times with DW and photographed using an Olympus IX71 microscope (Olympus, Tokyo, Japan). To quantify lipid accumulation, the stained cells were dissolved in 100% isopropanol and measured at 490 nm with a microplate reader (Bio-Rad).

### 4.7. Immunofluorescence Assay

The cells were treated with or without PC in 4-well confocal dishes. After treatment, the cells were fixed with 10% formalin for 10 min, treated with 0.2% Triton X in PBS for 20 min, washed three times with PBS, and then blocked with 5% BSA in PBS for 1 h. Next, the cells were incubated overnight with anti-SREBP1 antibody (1:500 in PBS) at 4 °C followed by Alexa Fluor-555 antibody for 1 h at room temperature. The cell nuclei were stained with 4,6-diamidino-2-phenylindole (DAPI; Sigma-Aldrich) for 3 min at room temperature. The cells stained with DAPI were washed with PBS for 30 min, and fluorescence was observed using a CELENA™ S Digital Imaging System (Logos Biosystems, Inc. Anyang-si, Gyeonggi-do, Korea).

### 4.8. Isolation of Total RNA and Real-Time Polymerase Chain Reaction (PCR)

Cells were treated with Ribo-Ex to extract RNA using the GeneAll Hybrid-R RNA Purification Kit (GeneAll, Seoul, Korea). RNA was quantified using NanoDrop (Thermo Fisher Scientific, Waltham, MA, USA). Amplification of complementary DNA was performed as follows: 45 °C for 60 min and 95 °C for 15 min at 4 °C using Maxime RT premix (iNtRON Biotechnology, Seongnam-si, Gyeonggi-do, Korea). Real-time quantitative PCR was performed using the Universal SYBR Green Master Mix (Applied Biosystems, Foster City, CA, USA). Real-time PCR was performed on the Applied Biosystems StepOne System (Applied Biosystems). In this study, relative target gene expression was quantified relative to that of glyceraldehyde 3-phosphate dehydrogenase to determine mRNA expression levels. The PCR primers used are shown in Table 1.

### 4.9. Statistical Analysis

Data are expressed as means ± standard errors of the means of at least three different experiments. Statistical analysis was performed using a one-way analysis of variance with GraphPad Prism v5 software (GraphPad Software, San Diego, CA, USA) followed by the Mann–Whitney test. A *p*-value < 0.05 was considered statistically significant, and *p* < 0.01 and *p* < 0.001 was considered highly significant.

## Figures and Tables

**Figure 1 ijms-22-10405-f001:**
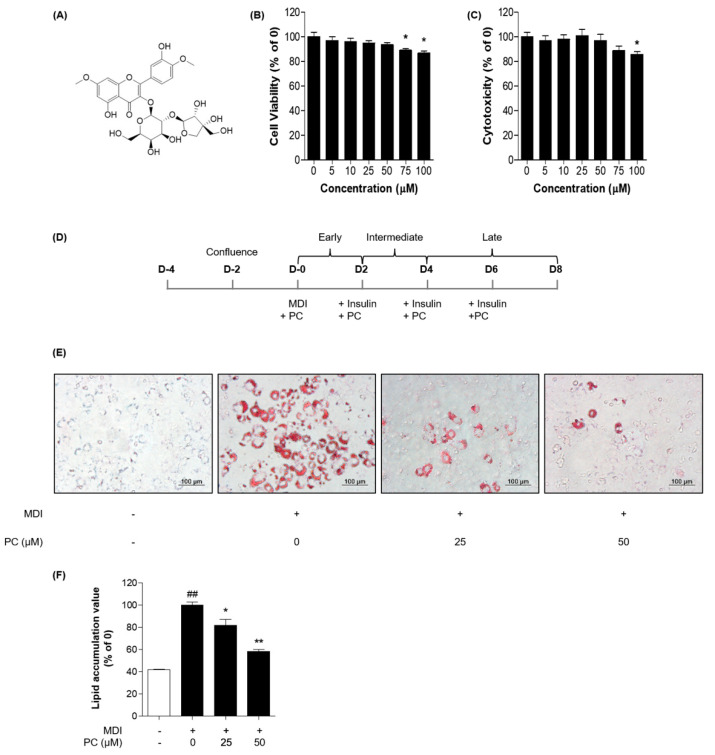
PC suppression of lipid accumulation in 3T3-L1 adipocytes: (**A**) Structure of PC; 3T3-L1 preadipocytes were treated with indicated concentrations of PC for 8 days, and cell viability was confirmed via the MTT assay (**B**) and LDH release assay (**C**); (**D**) Scheme of the 3T3-L1 cell differentiation experiment; 3T3-L1 cells were treated with PC for 8 days and then stained with Oil Red O. Lipid droplets were visualized (**E**) and quantified (**F**); The image data was photographed using an Olympus IX71 microscope (magnification: 200×, scale bar: 100 μm). The experimental data were obtained from at least three independent experiments and shown as means ± standard errors of the means. ## *p* < 0.01 compared to the control (non-treated) group. * *p* < 0.05 and ** *p* < 0.01 compared to the MDI-treated group.

**Figure 2 ijms-22-10405-f002:**
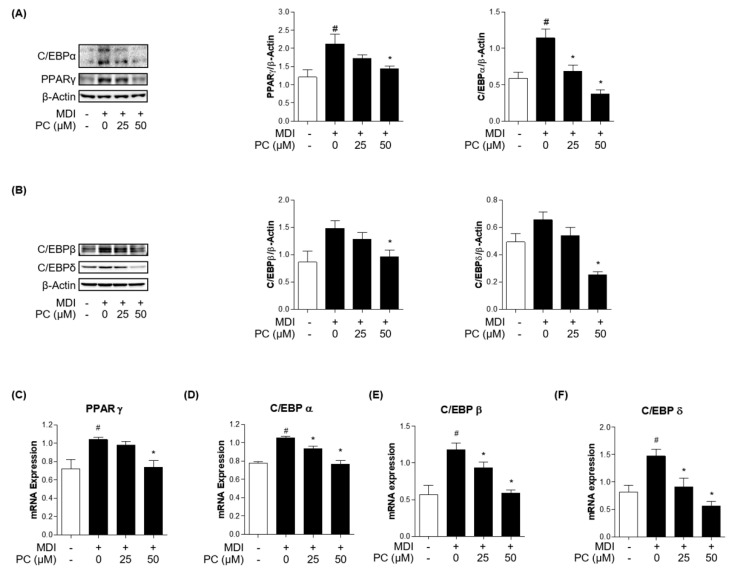
PC suppression of adipogenic factor expression levels in 3T3-L1 adipocytes: Cells were treated with MDI without or with PC. The protein levels of adipogenesis-related transcription factors were detected by Western blotting (**A**,**B**), and mRNA levels were detected by real-time PCR (**C**–**F**); The experimental data were obtained from at least three independent experiments and shown as means ± standard errors of the means. # *p* < 0.05 compared to the control (non-treated) group. * *p* < 0.05 compared to the MDI-treated group.

**Figure 3 ijms-22-10405-f003:**
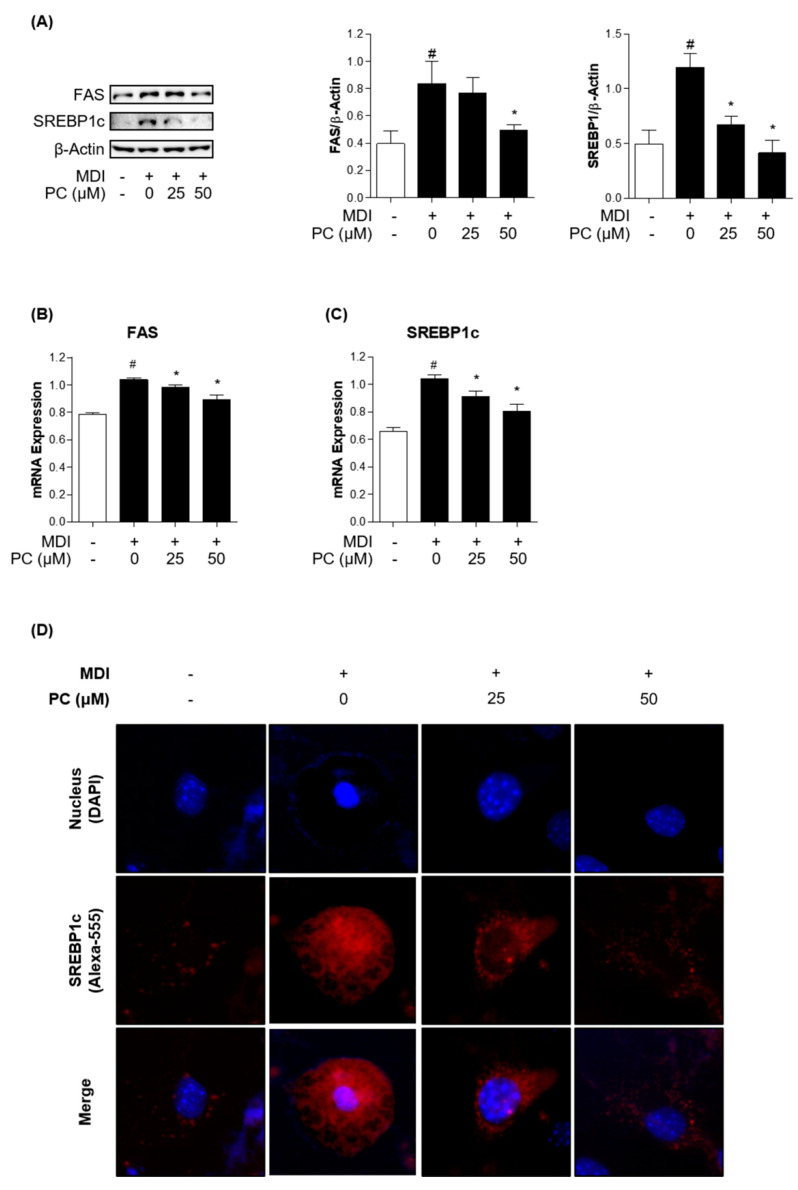
Effect of PC on lipogenic factor expression levels: The protein levels of FAS and SREBP1c were detected by Western blotting (**A**), and mRNA levels were detected by real-time PCR (**B**,**C**); (**D**) SREBP1c in differentiated 3T3-L1 adipocytes was detected by immunofluorescence analysis using microscopy (magnification: 400×). The experimental data were obtained from at least three independent experiments and shown as means ± standard errors of the means. # *p* < 0.05 compared to the control (non-treated) group. * *p* < 0.05 compared to the MDI-treated group.

**Figure 4 ijms-22-10405-f004:**
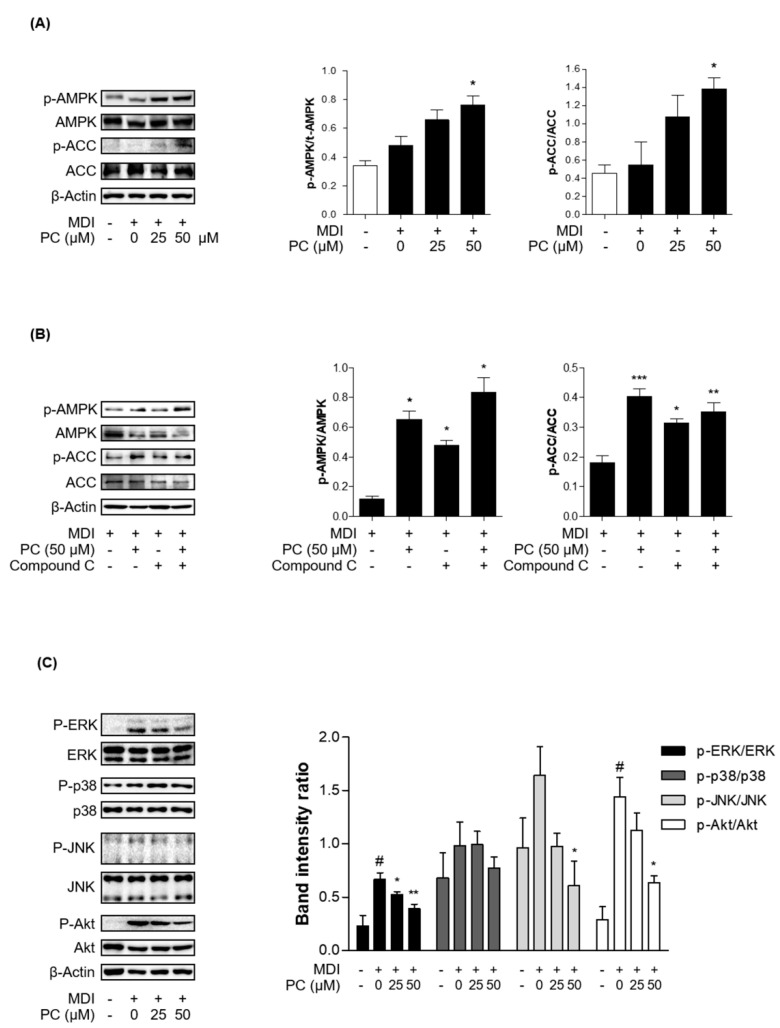
Effect of PC on AMPK/ACC and MAPK/Akt signaling pathway expression: (**A**) The protein levels of *p*-AMPK and *p*-ACC were detected by Western blotting. The ratios of *p*-AMPK/AMPK and *p*-ACC/ACC were determined using ImageJ software (National Institutes of Health, Bethesda, MD, USA); (**B**) AMPK activity was tested in the presence of compound C; (**C**) The protein levels of *p*-ERK, *p*-p38, *p*-JNK, and *p*-Akt were detected by Western blotting, and the ratios of *p*-ERK/ERK, *p*-p38/p38, *p*-JNK/JNK, and *p*-Akt/Akt were also determined using ImageJ software; The experimental data were obtained from at least three independent experiments and shown as means ± standard errors of the means. # *p* < 0.05 compared to the control (non-treated) group. * *p* < 0.05, ** *p* < 0.01 and *** *p* < 0.001 compared to the MDI-treated group.

**Figure 5 ijms-22-10405-f005:**
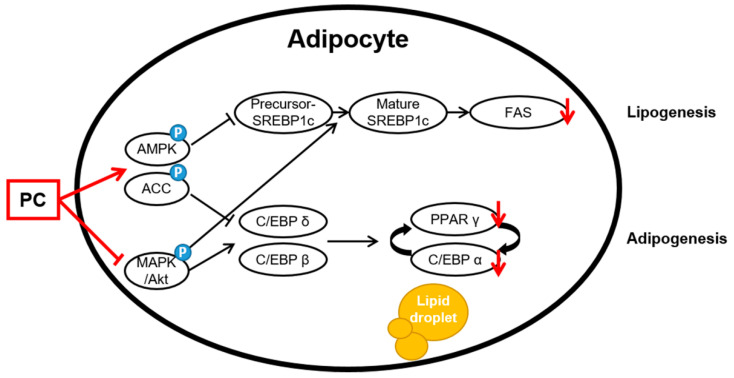
Schematic of the potential lipid mechanisms of PC in differentiated 3T3-L1 adipocytes. The anti-lipogenic and anti-adipogenic effects of PC are thought to be exerted via the suppression of the AMPK/ACC and MAPK/Akt pathways, thereby inhibiting adipogenic transcription factors and lipogenic proteins.

**Table 1 ijms-22-10405-t001:** Sequences of real-time PCR primers.

Genes	Primer	Forward Primers
*C/EBPβ*	Forward	AGCGGCTGCAGAAGAAGGT
	Reverse	GGCAGCTGCTTGAACAAGTTC
*C/EBPδ*	Forward	TTCCAACCCCTTCCCTGAT
	Reverse	CTGGAGGGTTTGTGTTTTCTGT
*PPARγ*	Forward	GGTGAAACTCTGGGAGATTC
	Reverse	CAACCATTGGGTCAGCTCTT
*C* */* *EBPα*	Forward	AGGTGCTGGAGTTGACCAGT
	Reverse	CAGCCTAGAGATCCAGCGAC
*FAS*	Forward	TTGCTGGCACTACAGAATGC
	Reverse	AACAGCCTCAGAGCGACAAT
*SREBP 1c*	Forward	ATCGCAAACAAGCTGACCTG
	Reverse	AGATCCAGGTTTGAGGTGGG
*GAPDH*	Forward	GCCACATCGCTCAGACACC
	Reverse	CCCAATACGACCAAATCCGT

## Data Availability

All data presented this study are available from the corresponding author, upon reasonable request.

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
