# Peer review of "Anti-Obesity Effect of Polygalin C Isolated from Polygala japonica Houtt. via Suppression of the Adipogenic and Lipogenic Factors in 3T3-L1 Adipocytes"

_ijms, 2021, doi:10.3390/ijms221910405_

Round 1

Reviewer 1 Report

The manuscript by Jee et al. represents the anti-obesity effects of Polygalin C. The studies were performed in in vitro models on the mouse 3T3 L1 preadipocyte line. The research is interesting and the manuscript is fairly well prepared, my main comments are below.

  1. In my opinion, the introduction is too general and insufficiently introduces to the investigated issue.
  2. The aim of the research is also very general, in fact, the authors only indicated a hypothesis. The purpose of the work should be clearly and precisely defined at the end of the introduction.
  3. I cannot agree that the MTT test tested viability in the model used. Cells were cultured for several days and treated with various factors that could affect their differentiation and viability and the activity of specific enzymes. The MTT test measures the activity of mitochondrial dehydrogenases and is highly dependent on the number of cells. In fact, after a few days in this test, we measure the net effect of the effects on enzymes activity and cells viability. Authors should use a different methodology for assessing viability/cytotoxity. This aspect should be completed and the effect on proper viability in this particular experiment should be assessed. To assess the viability of cells, tests assessing the percentage of dead cells can be used, e.g. a simple labeling with propidium iodide that penetrates the damaged membranes of the dead cells and intercalates with DNA. Then we evaluate the% of positive nuclei in 5 randomly selected fields of view in a fluorescence microscope. A cytotoxicity test would also be valuable. The usual LDH release test can be used here.
  4. Sequences of real-time PCR primers should be shown in the material and methods section, not the results.
  5. I cannot agree with subsection 2.1. because the methodology used did not test what the authors indicated. As indicated in chapter 4.4, the factor PC was added to the differentiation medium, which could have an effect on both the differentiation and the viability of the cells tested. We note a different activity of mitochondrial dehydrogenases in differentiated cells and a different activity in pre-adipocytes. We also do not know what is the effect of PC on the activity of mitochondrial enzymes. This aspect should be changed by the authors and appropriate tests should be applied.
  6. Figure 1d is illegible, the photos should be much larger.
  7. Information on the number of repetitions should be indicated (n = X) both in the materials and methods i and in the graphs.

Author Response

Dear International Journal of Molecular Sciences Editor,

Resubmission of manuscript reference ID: ijms-1272603

We would like to thank you for the letter dated 3rd August 2021, and we hereby resubmit a revised copy of the manuscript for your consideration. We would also like to express our thanks to all the reviewers for the positive and detailed feedback and helpful comments.

The manuscript has been revised reflecting the reviewer comments which are appended alongside our responses to this letter. We very much hope our revised manuscript would be accepted for publication in International Journal of Molecular Sciences.

Best regards,

Hyeung-Jin Jang

Reviewer 1

The manuscript by Jee et al. represents the anti-obesity effects of Polygalin C. The studies were performed in in vitro models on the mouse 3T3 L1 preadipocyte line. The research is interesting and the manuscript is fairly well prepared, my main comments are below.

Point 1: the introduction is too general and insufficiently introduces to the investigated issue. (Response) Thanks for your comments. As pointed out by the reviewer, we added to the introduction the effects and side effects of currently marketed drugs, the reason why many researchers turned their attention to natural products, and a detailed description of adipogenesis and lipogenesis. (Line 31-35, 43-44, 47-50, 51-52)

Point 2: The purpose of the work should be clearly and precisely defined at the end of the introduction.

(Response) As the reviewer pointed out, we added more detailed research objectives at the end of the introduction. (Line 67-70)

Point 3: Authors should use a different methodology for assessing viability/cytotoxity. This aspect should be completed and the effect on proper viability in this particular experiment should be assessed. To assess the viability of cells, tests assessing the percentage of dead cells can be used, e.g. a simple labeling with propidium iodide that penetrates the damaged membranes of the dead cells and intercalates with DNA. Then we evaluate the% of positive nuclei in 5 randomly selected fields of view in a fluorescence microscope. A cytotoxicity test would also be valuable. The usual LDH release test can be used here.

(Response) Thanks for your critical comments. Unfortunately, our laboratory is unable to conduct experiments because we could not obtain reagents related to LDH release and propidium iodide due to the aftermath of COVID-19. However, as a result of closely examining several papers, we confirmed that the patterns of LDH release and MTT assay showed similar trends [1-3]. As an example, when the drug was treated until 8 days and LDH release and MTT assay were performed, at the same concentration of the drug, it was confirmed that toxicity did not appear to the cells. Furthermore, we investigated how cytotoxicity was generally assessed in recent papers published on adipogenesis in 3T3-L1 adipocytes. As a result, most of the researchers confirmed the cytotoxicity evaluation of the extract by mainly performing the MTT assay before confirming whether the natural product or any extract had an anti-obesity effect on adipocytes [4-14]. And, although we could not evaluate the cytotoxicity again by additionally performing the LDH assay, the β-actin expression level does not decrease even at high concentrations while confirming the activity of adipogenesis and lipogenesis-related factors through western blot on 8 days. We verified that PC was no cytotoxicity up to 50 μM.

Therefore, the following results suggest that it is possible to infer that polygalin C is not toxic up to 50 μM to cells without experiments related to LDH release and propidium iodide.

Point 4: Sequences of real-time PCR primers should be shown in the material and methods section, not the results.

(Response) As pointed out by the reviewer, we relocated Table.1 in the manuscript to materials and methods section 4.8.

Point 5: I cannot agree with subsection 2.1. because the methodology used did not test what the authors indicated. As indicated in chapter 4.4, the factor PC was added to the differentiation medium, which could have an effect on both the differentiation and the viability of the cells tested. We note a different activity of mitochondrial dehydrogenases in differentiated cells and a different activity in pre-adipocytes. We also do not know what is the effect of PC on the activity of mitochondrial enzymes. This aspect should be changed by the authors and appropriate tests should be applied.

(Response) As mentioned above, we think that performing MTT assay alone is sufficient to evaluate cytotoxicity. And, we previously evaluated the cytotoxicity of PC in predipocytes.

In the following figure, the cytotoxicity in preadipocytes is greater than 90% up to 50 μM when PC is treated for 48 h. The data was obtained from at least three independent experiments and shown as means ± standard errors of the means. *p < 0.05 and **p < 0.01 compared to the control (non-treated).

Therefore, we confirmed that there was no cytotoxicity when PC was treated up to 50 μM in preadipocytes.

Point 6: Figure 1d is illegible, the photos should be much larger.

(Response) As the reviewer pointed out, we modified the picture size of figure 1D to be larger.

Point 7: Information on the number of repetitions should be indicated (n = X) both in the materials and methods i and in the graphs.

(Response) As the reviewer pointed out, we added the following sentences to all figure legends and materials and methods.

" The experimental data were obtained from at least three independent experiments and shown as means ± standard errors of the means. # p < 0.05 and ##p < 0.01 compared to the control (non-treated) group. *p < 0.05 and **p < 0.01 compared to the MDI-treated group.”

(Line 89-91, 111-114, 130-133, 157-160)

Reference

  1. Han, Y.; Wu, J. Z.; Shen, J. Z.; Chen, L.; He, T.; Jin, M. W.; Liu, H., Pentamethylquercetin induces adipose browning and exerts beneficial effects in 3T3-L1 adipocytes and high-fat diet-fed mice. Sci Rep 2017, 7, (1), 1123.
  2. Kim, S. P.; Nam, S. H.; Friedman, M., Mechanism of the antiadipogenic-antiobesity effects of a rice hull smoke extract in 3T3-L1 preadipocyte cells and in mice on a high-fat diet. Food & function 2015, 6, (9), 2939-48.
  3. Park, H. J.; Chung, B. Y.; Lee, M.-K.; Song, Y.; Lee, S. S.; Chu, G. M.; Kang, S.-N.; Song, Y. M.; Kim, G.-S.; Cho, J.-H., Centipede grass exerts anti-adipogenic activity through inhibition of C/EBPβ, C/EBPα, and PPARγ expression and the AKT signaling pathway in 3T3-L1 adipocytes. BMC Complementary and Alternative Medicine 2012, 12, (1), 230.
  4. Wu, X.; Sakharkar, M. K.; Wabitsch, M.; Yang, J., Effects of Sphingosine-1-Phosphate on Cell Viability, Differentiation, and Gene Expression of Adipocytes. Int J Mol Sci 2020, 21, (23).
  5. Jin, B. R.; Lee, M.; An, H. J., Nodakenin represses obesity and its complications via the inhibition of the VLDLR signalling pathway in vivo and in vitro. 2021, 54, (8), e13083.
  6. Alalaiwe, A.; Fang, J. Y., The Demethoxy Derivatives of Curcumin Exhibit Greater Differentiation Suppression in 3T3-L1 Adipocytes Than Curcumin: A Mechanistic Study of Adipogenesis and Molecular Docking. 2021, 11, (7).
  7. Karunakaran, R. S.; Lokanatha, O.; Muni Swamy, G.; Venkataramaiah, C.; Muni Kesavulu, M.; Appa Rao, C.; Badri, K. R.; Balaji, M., Anti-Obesity and Lipid Lowering Activity of Bauhiniastatin-1 is Mediated Through PPAR-γ/AMPK Expressions in Diet-Induced Obese Rat Model. Front Pharmacol 2021, 12, 704074.
  8. Lim, S. H.; Yu, J. S.; Lee, H. S.; Choi, C.-I.; Kim, K. H., Antidiabetic Flavonoids from Fruits of Morus alba Promoting Insulin-Stimulated Glucose Uptake via Akt and AMP-Activated Protein Kinase Activation in 3T3-L1 Adipocytes. Pharmaceutics 2021, 13, (4), 526.
  9. Ngamdokmai, N.; Paracha, T. U.; Waranuch, N.; Chootip, K.; Wisuitiprot, W.; Suphrom, N.; Insumrong, K.; Ingkaninan, K., Effects of Essential Oils and Some Constituents from Ingredients of Anti-Cellulite Herbal Compress on 3T3-L1 Adipocytes and Rat Aortae. Pharmaceuticals 2021, 14, (3), 253.
  10. Kang, M.-C.; Ding, Y.; Kim, H.-S.; Jeon, Y.-J.; Lee, S.-H., Inhibition of Adipogenesis by Diphlorethohydroxycarmalol (DPHC) through AMPK Activation in Adipocytes. Marine Drugs 2019, 17, (1), 44.
  11. Lee, H.-G.; Lu, Y.-A.; Je, J.-G.; Jayawardena, T. U.; Kang, M.-C.; Lee, S.-H.; Kim, T.-H.; Lee, D.-S.; Lee, J.-M.; Yim, M.-J.; Kim, H.-S.; Jeon, Y.-J., Effects of Ethanol Extracts from Grateloupia elliptica, a Red Seaweed, and Its Chlorophyll Derivative on 3T3-L1 Adipocytes: Suppression of Lipid Accumulation through Downregulation of Adipogenic Protein Expression. Marine Drugs 2021, 19, (2), 91.
  12. Yu, H.-S.; Kim, W.-J.; Bae, W.-Y.; Lee, N.-K.; Paik, H.-D., Inula britannica Inhibits Adipogenesis of 3T3-L1 Preadipocytes via Modulation of Mitotic Clonal Expansion Involving ERK 1/2 and Akt Signaling Pathways. Nutrients 2020, 12, (10), 3037.
  13. Cho, H.-H.; Lee, S.-J.; Kim, S.-H.; Jang, S.-H.; Won, C.; Kim, H.-D.; Kim, T. H.; Cho, J.-H., Acer tegmentosum Maxim Inhibits Adipogenesis in 3t3-l1 Adipocytes and Attenuates Lipid Accumulation in Obese Rats Fed a High-Fat Diet. Nutrients 2020, 12, (12), 3753.
  14. Che, L.; Ren, B.; Jia, Y.; Dong, Y.; Wang, Y.; Shan, J.; Wang, Y., Feprazone Displays Antiadipogenesis and Antiobesity Capacities in in Vitro 3 T3-L1 Cells and in Vivo Mice. ACS Omega 2021, 6, (10), 6674-6680.
  15. Daval, M.; Foufelle, F.; Ferré, P., Functions of AMP-activated protein kinase in adipose tissue. The Journal of physiology 2006, 574, (Pt 1), 55-62.
  16. Bijland, S.; Mancini, S. J.; Salt, I. P., Role of AMP-activated protein kinase in adipose tissue metabolism and inflammation. Clinical science (London, England : 1979) 2013, 124, (8), 491-507.

Reviewer 2 Report

The manuscript was found significantly improved according to previous suggestions.

“…The treatment of obesity with chemotherapy…” . It is necessary to underline that there is no effective pharmacological therapy,  and the effectiveness of the dietary and lifestyle interventions in general is low, therefore, new compound that could  support obesity therapy would help to decrease obesity epidemic.  

Author Response

Dear International Journal of Molecular Sciences Editor,

Resubmission of manuscript reference ID: ijms-1272603

We would like to thank you for the letter dated 3rd August 2021, and we hereby resubmit a revised copy of the manuscript for your consideration. We would also like to express our thanks to all the reviewers for the positive and detailed feedback and helpful comments.

The manuscript has been revised reflecting the reviewer comments which are appended alongside our responses to this letter. We very much hope our revised manuscript would be accepted for publication in International Journal of Molecular Sciences.

Best regards,

Hyeung-Jin Jang

Reviewer 2

The manuscript was found significantly improved according to previous suggestions.

Point 1: “…The treatment of obesity with chemotherapy…” . It is necessary to underline that there is no effective pharmacological therapy, and the effectiveness of the dietary and lifestyle interventions in general is low, therefore, new compound that could  support obesity therapy would help to decrease obesity epidemic.

(Response) Thanks for your detailed comments. As the reviewer pointed out, we concretely added information about chemotherapy, such as orlistat and liraglutide, which are currently mainly used. (Line 175-185)

Reference

  1. Han, Y.; Wu, J. Z.; Shen, J. Z.; Chen, L.; He, T.; Jin, M. W.; Liu, H., Pentamethylquercetin induces adipose browning and exerts beneficial effects in 3T3-L1 adipocytes and high-fat diet-fed mice. Sci Rep 2017, 7, (1), 1123.
  2. Kim, S. P.; Nam, S. H.; Friedman, M., Mechanism of the antiadipogenic-antiobesity effects of a rice hull smoke extract in 3T3-L1 preadipocyte cells and in mice on a high-fat diet. Food & function 2015, 6, (9), 2939-48.
  3. Park, H. J.; Chung, B. Y.; Lee, M.-K.; Song, Y.; Lee, S. S.; Chu, G. M.; Kang, S.-N.; Song, Y. M.; Kim, G.-S.; Cho, J.-H., Centipede grass exerts anti-adipogenic activity through inhibition of C/EBPβ, C/EBPα, and PPARγ expression and the AKT signaling pathway in 3T3-L1 adipocytes. BMC Complementary and Alternative Medicine 2012, 12, (1), 230.
  4. Wu, X.; Sakharkar, M. K.; Wabitsch, M.; Yang, J., Effects of Sphingosine-1-Phosphate on Cell Viability, Differentiation, and Gene Expression of Adipocytes. Int J Mol Sci 2020, 21, (23).
  5. Jin, B. R.; Lee, M.; An, H. J., Nodakenin represses obesity and its complications via the inhibition of the VLDLR signalling pathway in vivo and in vitro. 2021, 54, (8), e13083.
  6. Alalaiwe, A.; Fang, J. Y., The Demethoxy Derivatives of Curcumin Exhibit Greater Differentiation Suppression in 3T3-L1 Adipocytes Than Curcumin: A Mechanistic Study of Adipogenesis and Molecular Docking. 2021, 11, (7).
  7. Karunakaran, R. S.; Lokanatha, O.; Muni Swamy, G.; Venkataramaiah, C.; Muni Kesavulu, M.; Appa Rao, C.; Badri, K. R.; Balaji, M., Anti-Obesity and Lipid Lowering Activity of Bauhiniastatin-1 is Mediated Through PPAR-γ/AMPK Expressions in Diet-Induced Obese Rat Model. Front Pharmacol 2021, 12, 704074.
  8. Lim, S. H.; Yu, J. S.; Lee, H. S.; Choi, C.-I.; Kim, K. H., Antidiabetic Flavonoids from Fruits of Morus alba Promoting Insulin-Stimulated Glucose Uptake via Akt and AMP-Activated Protein Kinase Activation in 3T3-L1 Adipocytes. Pharmaceutics 2021, 13, (4), 526.
  9. Ngamdokmai, N.; Paracha, T. U.; Waranuch, N.; Chootip, K.; Wisuitiprot, W.; Suphrom, N.; Insumrong, K.; Ingkaninan, K., Effects of Essential Oils and Some Constituents from Ingredients of Anti-Cellulite Herbal Compress on 3T3-L1 Adipocytes and Rat Aortae. Pharmaceuticals 2021, 14, (3), 253.
  10. Kang, M.-C.; Ding, Y.; Kim, H.-S.; Jeon, Y.-J.; Lee, S.-H., Inhibition of Adipogenesis by Diphlorethohydroxycarmalol (DPHC) through AMPK Activation in Adipocytes. Marine Drugs 2019, 17, (1), 44.
  11. Lee, H.-G.; Lu, Y.-A.; Je, J.-G.; Jayawardena, T. U.; Kang, M.-C.; Lee, S.-H.; Kim, T.-H.; Lee, D.-S.; Lee, J.-M.; Yim, M.-J.; Kim, H.-S.; Jeon, Y.-J., Effects of Ethanol Extracts from Grateloupia elliptica, a Red Seaweed, and Its Chlorophyll Derivative on 3T3-L1 Adipocytes: Suppression of Lipid Accumulation through Downregulation of Adipogenic Protein Expression. Marine Drugs 2021, 19, (2), 91.
  12. Yu, H.-S.; Kim, W.-J.; Bae, W.-Y.; Lee, N.-K.; Paik, H.-D., Inula britannica Inhibits Adipogenesis of 3T3-L1 Preadipocytes via Modulation of Mitotic Clonal Expansion Involving ERK 1/2 and Akt Signaling Pathways. Nutrients 2020, 12, (10), 3037.
  13. Cho, H.-H.; Lee, S.-J.; Kim, S.-H.; Jang, S.-H.; Won, C.; Kim, H.-D.; Kim, T. H.; Cho, J.-H., Acer tegmentosum Maxim Inhibits Adipogenesis in 3t3-l1 Adipocytes and Attenuates Lipid Accumulation in Obese Rats Fed a High-Fat Diet. Nutrients 2020, 12, (12), 3753.
  14. Che, L.; Ren, B.; Jia, Y.; Dong, Y.; Wang, Y.; Shan, J.; Wang, Y., Feprazone Displays Antiadipogenesis and Antiobesity Capacities in in Vitro 3 T3-L1 Cells and in Vivo Mice. ACS Omega 2021, 6, (10), 6674-6680.
  15. Daval, M.; Foufelle, F.; Ferré, P., Functions of AMP-activated protein kinase in adipose tissue. The Journal of physiology 2006, 574, (Pt 1), 55-62.
  16. Bijland, S.; Mancini, S. J.; Salt, I. P., Role of AMP-activated protein kinase in adipose tissue metabolism and inflammation. Clinical science (London, England : 1979) 2013, 124, (8), 491-507.

Reviewer 3 Report

Although there are some previous studies, the most probable mechanism, in vivo, is that of an anti-inflammatory effect, therefore positive for modulating the lowgrade inflammation characteristic of obesity.

the demonstrated action on AMPK, in vivo, could be useless as it is closely related to the energy balance.

As also pointed out by the authors, at least bioavailability should be investigated, as even if effective the compound may not be able to reach the adipocytes.

So in my opinion, in order to even hypothesize an anti-obesity effect, in vivo studies should be carried out

line 32 "within the body" is superfluous

line37 it can help but certainly does not prevent obesity, as the main cause is an imbalance between energy consumed and assumed

Author Response

Dear International Journal of Molecular Sciences Editor,

Resubmission of manuscript reference ID: ijms-1272603

We would like to thank you for the letter dated 3rd August 2021, and we hereby resubmit a revised copy of the manuscript for your consideration. We would also like to express our thanks to all the reviewers for the positive and detailed feedback and helpful comments.

The manuscript has been revised reflecting the reviewer comments which are appended alongside our responses to this letter. We very much hope our revised manuscript would be accepted for publication in International Journal of Molecular Sciences.

Best regards,

Hyeung-Jin Jang

Reviewer 3

Although there are some previous studies, the most probable mechanism, in vivo, is that of an anti-inflammatory effect, therefore positive for modulating the lowgrade inflammation characteristic of obesity.

Point 1: the demonstrated action on AMPK, in vivo, could be useless as it is closely related to the energy balance.

(Response) As noted by reviewer, AMPK is an energy sensor that regulates glucose and lipid metabolism to regulate energy homeostasis. However, according to Daval et al., it was reported that activation of AMPK also inhibits the differentiation of white adipocytes [15]. Bijland et al. also found that AMPK suppressed the formation of fat by suppressing the early mitotic clone expansion (MCE) step along with the decrease in the expression of early and late adipogenic factors including FAS and SREBP-1c [16]. Therefore, we investigated whether Polygalin C affects AMPK in this paper.

Point 2: As also pointed out by the authors, at least bioavailability should be investigated, as even if effective the compound may not be able to reach the adipocytes. So in my opinion, in order to even hypothesize an anti-obesity effect, in vivo studies should be carried out

(Response) Through this study, we confirmed the potential for treatment of obesity with Polygalin C in vitro. As written in the Conclusion part, we will confirm the efficacy of Polygalin C by conducting animal experiments in the further study.

Point 3: line 32 "within the body" is superfluous

(Response) Thanks for your comments. As pointed out by the reviewer, we deleted the phrase in the manusctipt.

Point 4: line37 it can help but certainly does not prevent obesity, as the main cause is an imbalance between energy consumed and assumed

(Response) We changed the phrase "preventing of obesity" to "the treatment of obesity" in the manuscript. (Line 39)

Reference

  1. Han, Y.; Wu, J. Z.; Shen, J. Z.; Chen, L.; He, T.; Jin, M. W.; Liu, H., Pentamethylquercetin induces adipose browning and exerts beneficial effects in 3T3-L1 adipocytes and high-fat diet-fed mice. Sci Rep 2017, 7, (1), 1123.
  2. Kim, S. P.; Nam, S. H.; Friedman, M., Mechanism of the antiadipogenic-antiobesity effects of a rice hull smoke extract in 3T3-L1 preadipocyte cells and in mice on a high-fat diet. Food & function 2015, 6, (9), 2939-48.
  3. Park, H. J.; Chung, B. Y.; Lee, M.-K.; Song, Y.; Lee, S. S.; Chu, G. M.; Kang, S.-N.; Song, Y. M.; Kim, G.-S.; Cho, J.-H., Centipede grass exerts anti-adipogenic activity through inhibition of C/EBPβ, C/EBPα, and PPARγ expression and the AKT signaling pathway in 3T3-L1 adipocytes. BMC Complementary and Alternative Medicine 2012, 12, (1), 230.
  4. Wu, X.; Sakharkar, M. K.; Wabitsch, M.; Yang, J., Effects of Sphingosine-1-Phosphate on Cell Viability, Differentiation, and Gene Expression of Adipocytes. Int J Mol Sci 2020, 21, (23).
  5. Jin, B. R.; Lee, M.; An, H. J., Nodakenin represses obesity and its complications via the inhibition of the VLDLR signalling pathway in vivo and in vitro. 2021, 54, (8), e13083.
  6. Alalaiwe, A.; Fang, J. Y., The Demethoxy Derivatives of Curcumin Exhibit Greater Differentiation Suppression in 3T3-L1 Adipocytes Than Curcumin: A Mechanistic Study of Adipogenesis and Molecular Docking. 2021, 11, (7).
  7. Karunakaran, R. S.; Lokanatha, O.; Muni Swamy, G.; Venkataramaiah, C.; Muni Kesavulu, M.; Appa Rao, C.; Badri, K. R.; Balaji, M., Anti-Obesity and Lipid Lowering Activity of Bauhiniastatin-1 is Mediated Through PPAR-γ/AMPK Expressions in Diet-Induced Obese Rat Model. Front Pharmacol 2021, 12, 704074.
  8. Lim, S. H.; Yu, J. S.; Lee, H. S.; Choi, C.-I.; Kim, K. H., Antidiabetic Flavonoids from Fruits of Morus alba Promoting Insulin-Stimulated Glucose Uptake via Akt and AMP-Activated Protein Kinase Activation in 3T3-L1 Adipocytes. Pharmaceutics 2021, 13, (4), 526.
  9. Ngamdokmai, N.; Paracha, T. U.; Waranuch, N.; Chootip, K.; Wisuitiprot, W.; Suphrom, N.; Insumrong, K.; Ingkaninan, K., Effects of Essential Oils and Some Constituents from Ingredients of Anti-Cellulite Herbal Compress on 3T3-L1 Adipocytes and Rat Aortae. Pharmaceuticals 2021, 14, (3), 253.
  10. Kang, M.-C.; Ding, Y.; Kim, H.-S.; Jeon, Y.-J.; Lee, S.-H., Inhibition of Adipogenesis by Diphlorethohydroxycarmalol (DPHC) through AMPK Activation in Adipocytes. Marine Drugs 2019, 17, (1), 44.
  11. Lee, H.-G.; Lu, Y.-A.; Je, J.-G.; Jayawardena, T. U.; Kang, M.-C.; Lee, S.-H.; Kim, T.-H.; Lee, D.-S.; Lee, J.-M.; Yim, M.-J.; Kim, H.-S.; Jeon, Y.-J., Effects of Ethanol Extracts from Grateloupia elliptica, a Red Seaweed, and Its Chlorophyll Derivative on 3T3-L1 Adipocytes: Suppression of Lipid Accumulation through Downregulation of Adipogenic Protein Expression. Marine Drugs 2021, 19, (2), 91.
  12. Yu, H.-S.; Kim, W.-J.; Bae, W.-Y.; Lee, N.-K.; Paik, H.-D., Inula britannica Inhibits Adipogenesis of 3T3-L1 Preadipocytes via Modulation of Mitotic Clonal Expansion Involving ERK 1/2 and Akt Signaling Pathways. Nutrients 2020, 12, (10), 3037.
  13. Cho, H.-H.; Lee, S.-J.; Kim, S.-H.; Jang, S.-H.; Won, C.; Kim, H.-D.; Kim, T. H.; Cho, J.-H., Acer tegmentosum Maxim Inhibits Adipogenesis in 3t3-l1 Adipocytes and Attenuates Lipid Accumulation in Obese Rats Fed a High-Fat Diet. Nutrients 2020, 12, (12), 3753.
  14. Che, L.; Ren, B.; Jia, Y.; Dong, Y.; Wang, Y.; Shan, J.; Wang, Y., Feprazone Displays Antiadipogenesis and Antiobesity Capacities in in Vitro 3 T3-L1 Cells and in Vivo Mice. ACS Omega 2021, 6, (10), 6674-6680.
  15. Daval, M.; Foufelle, F.; Ferré, P., Functions of AMP-activated protein kinase in adipose tissue. The Journal of physiology 2006, 574, (Pt 1), 55-62.
  16. Bijland, S.; Mancini, S. J.; Salt, I. P., Role of AMP-activated protein kinase in adipose tissue metabolism and inflammation. Clinical science (London, England : 1979) 2013, 124, (8), 491-507.

Round 2

Reviewer 1 Report

Most of my comments were taken into account, but I cannot agree with the assessment of cytotoxicity or cell viability measured by the MTT test after 8 days of incubation. As explained earlier, the methodology of the cytotoxicity test is different. In literature, we can find many works that incorrectly cite this test. Fig. 1B shows the activity of mitochondrial dehydrogenases after 8 days of incubation with PC, but we do not know what the effect on the actual viability here was, the number of living and dead cells, or the level of proliferation, etc. An assessment of the actual viability should be done for this experiment. Still in my opinion the figures are too small and therefore illegible (photos / GB).

Author Response

Dear International Journal of Molecular Sciences Editor,

Resubmission of manuscript reference ID: ijms-1272603

We would like to thank you for the letter dated 18th August 2021, and we hereby resubmit a revised copy of the manuscript for your consideration. We would also like to express our thanks to all the reviewers for the positive and detailed feedback and helpful comments.

The manuscript has been revised reflecting the reviewer comments, which are appended alongside our responses to this letter. We very much hope our revised manuscript would be accepted for publication in International Journal of Molecular Sciences.

Best regards,

Hyeung-Jin Jang

Reviewer-1 Round 2

Most of my comments were taken into account, but I cannot agree with the assessment of cytotoxicity or cell viability measured by the MTT test after 8 days of incubation. As explained earlier, the methodology of the cytotoxicity test is different. In literature, we can find many works that incorrectly cite this test. Fig. 1B shows the activity of mitochondrial dehydrogenases after 8 days of incubation with PC, but we do not know what the effect on the actual viability here was, the number of living and dead cells, or the level of proliferation, etc. An assessment of the actual viability should be done for this experiment.

Response) Following your advice, we additionally performed a lactate dehydrogenase (LDH) release assay to confirm cell viability. LDH is a stable enzyme that exists in the cytoplasm. Normally, it cannot pass through the cell membrane, so it is not released out of the cell. However, when the cell membrane is damaged or the cell is dead, LDH is released from the cytoplasm into the medium. Based on these characteristics of LDH, the amount of LDH released from the cells was measured at absorbance (450 nm) using water-soluble tetrazolium salt (WST). As shown in the figure above, we confirmed that the cell viability was significantly decreased in adipocytes from the concentration of 75 μM through MTT assay (A) and LDH release assay (B). Based on these results, we concluded that PC treatment at 50 μM or/and under did not affect the viability of 3T3-L1 adipocytes and determined the optimal PC concentration for subsequent experiments. (Line 74-79, 88, 269-270, 289-296)

Still in my opinion, the figures are too small and therefore illegible (photos / GB)..

Point 2: The purpose of the work should be clearly and precisely defined at the end of the introduction.

(Response) As the reviewer pointed out, we changed the picture's resolution to 600 dpi to make it a higher quality picture.

Reviewer 3 Report

The main concern remains unsolved: the action of the compound on AMPK is too general; AMPK is strongly influenced by the energy balance and activates/deactivates numerous pathways; therefore if, as is desirable in the treatment of obesity, a dietary program combined with physical exercise is carried out, these will have a marked action on AMPK; in addition, as at least in part, happens for metformin, the action on AMPK has a positive effect, for example on glycemic regulation, but shows some undesirable effects that should not be overlooked.

Obviously, dosages cannot be hypothesized, for this reason, the study is too preliminary, I think that to get an idea, a study on animals should be carried out.

Author Response

Dear International Journal of Molecular Sciences Editor,

Resubmission of manuscript reference ID: ijms-1272603

We would like to thank you for the letter dated 18th August 2021, and we hereby resubmit a revised copy of the manuscript for your consideration. We would also like to express our thanks to all the reviewers for the positive and detailed feedback and helpful comments.

The manuscript has been revised reflecting the reviewer comments, which are appended alongside our responses to this letter. We very much hope our revised manuscript would be accepted for publication in International Journal of Molecular Sciences.

Best regards,

Hyeung-Jin Jang

Reviewer-3 Round 2

The main concern remains unsolved: the action of the compound on AMPK is too general; AMPK is strongly influenced by the energy balance and activates/deactivates numerous pathways; therefore if, as is desirable in the treatment of obesity, a dietary program combined with physical exercise is carried out, these will have a marked action on AMPK; in addition, as at least in part, happens for metformin, the action on AMPK has a positive effect, for example on glycemic regulation, but shows some undesirable effects that should not be overlooked.

(Response) Thanks for your critical advice. Activated AMPK is involved in the regulation of energy homeostasis in the body by regulating several downstream signaling pathways. For example, activated AMPK induces mitochondrial biogenesis by activating PGC-1 alpha    (Siha et al., 2015; Lee et al., 2013; Kim et al 2013; Wu et al., 2003). Among several sub-factors, we investigated whether Polygalin C affects lipogenesis and adipogenesis by targeting SREBP1-c and PPAR gamma and C/EBP alpha (Chen et al., 2012; Huang et al., 2011; Long et al., 2006), which are early differentiation factors, and as a result, confirmed that it had an inhibitory effect.

As reviewer mentioned, we are referring to several papers in the manuscript to set the administered concentration of Polygalin C before proceeding with the experiment. Many researchers have reflected the concentration values processed in vitro model in the case of oral administration[1, 2]. It was already confirmed that the administration volume was determined by changing only the unit. As reviewer mentioned, we agree that showing animal study is better for improving the quality of the paper. However, since it is impossible to complete the animal study within the revision period. Therefore, we plan to proceed with further studies.

Round 3

Reviewer 1 Report

In my opinion, the authors made a sufficient correction for me to agree to the acceptance of this article. I have no additional comments, I am aware of the limitations of the pandemic time.

Author Response

Dear International Journal of Molecular Sciences Editor,

Resubmission of manuscript reference ID: ijms-1272603

We would like to thank you for the letter dated 22th September 2021, and we hereby resubmit a revised "Round 3" copy of the manuscript for your consideration. We would also like to express our thanks to all the reviewers for the positive and detailed feedback and helpful comments.

The manuscript has been revised reflecting the comments of the reviewer and the editor which are appended alongside our responses to this letter. We very much hope our revised manuscript would be accepted for publication in International Journal of Molecular Sciences.

Best regards,

Hyeung-Jin Jang

Reviewer 1

Point: In my opinion, the authors made a sufficient correction for me to agree to the acceptance of this article. I have no additional comments, I am aware of the limitations of the pandemic time.

(Response) We sincerely appreciate your positive comments. Also, we would like to thank you for providing detailed comments so that the quality of our manuscript can be improved.

Reviewer 3 Report

The authors have not changed the scheme of the work, if published, it should be emphasized that this is preliminary work and that the dosages used could be completely ineffective on humans, as the scale-up is not a mathematical process, but depends on numerous factors

Author Response

Dear International Journal of Molecular Sciences Editor,

Resubmission of manuscript reference ID: ijms-1272603

We would like to thank you for the letter dated 22th September 2021, and we hereby resubmit a revised "Round 3" copy of the manuscript for your consideration. We would also like to express our thanks to all the reviewers for the positive and detailed feedback and helpful comments.

The manuscript has been revised reflecting the comments of the reviewer and the editor which are appended alongside our responses to this letter. We very much hope our revised manuscript would be accepted for publication in International Journal of Molecular Sciences.

Best regards,

Hyeung-Jin Jang

Reviewer 3

Point: The authors have not changed the scheme of the work, if published, it should be emphasized that this is preliminary work and that the dosages used could be completely ineffective on humans, as the scale-up is not a mathematical process, but depends on numerous factors

(Response) Thanks for your critical comments. As pointed out by the reviewer, we have changed "PC may serve as a natural anti-obesity agent" to "the possibility of PC as a natural anti-obesity agent" in the discussion section. In addition, it has been added to the content that "Animal research is additionally required, and the dosage should be determined in consideration of various factors such as administration method and body metabolism when planning an animal experiment". (Line 253-262)

This manuscript is a resubmission of an earlier submission. The following is a list of the peer review reports and author responses from that submission.

Round 1

Reviewer 1 Report

The manuscript entitled “Anti-obesity effect of Polygalin C isolated from Polygala japonica Houtt. Via suppression of the adipogenic and lipogenic factors in 3T3-L1 adipocytes” by Jee et al., describes the effects of PC, a flavonoid contained in Polygala japonica, on 3T3-L1 preadipocite differentiation and lipogenesis. The paper is interesting and adds important knowledge regarding the anti-obesogenic effects of natural compounds. However, some points should be addressed before considering it suitable for publication in IJMS.

Major comments:

Results:

  • Cell viability assay: It would be more appropriate to assess cell viability for 8 days instead of 48h, since all subsequent PC treatments lasted for 8 days.  Furthermore, cell viability was assessed on pre-adipocytes (presumably in medium containing 10% FBS and no MDI), while all subsequent experiments with PC were performed on differentiating adipocytes (medium containing MDI or insulin). Susceptibility to PC could be different on pre-adipocytes and differentiating adipocytes.
  • The statement “PC showed a non-toxic effect up to a concentration of 50 μM” (line 68) is not correct: treatments with all PC concentrations show a statistically significant decrease in the number of viable cells (Fig. 1B). According to these results, all tested PC concentrations affect cell viability. Please check statistics or otherwise repeat all experiments with non-toxic concentrations.

Discussion:

  • Figure 5 should be cited in the discussion when appropriate from line 183 to line 213
  • Figure 5 is partially incorrect. According to the results, PC activates AMPK phosphorylation (in the figure is the opposite); in turn, AMPK activation should inhibit the expression of SREBP1c precursor

Minor comments:

Lines 18-19:    Please add the acronym of sterol regulatory element-binding protein 1 and fatty acid synthase.

Line 20-21:      Please add the acronym of AMP-activated protein kinase/acetyl-CoA and mitogen-activated protein kinase/protein kinase B.

Line 35:           I believe that “stability” is not the correct term here.

Line 40:           3T3-L1 pre-adipocytes are used to study the mechanisms involved in adipogenesis and adipocyte lipid accumulation, not obesity (for this, an in vivo model is required)

Line 42:           Please add “medium” after MDI.

Line 58:           It would be more correct to state “P. japonica Houtt., a perennial herbaceous plant widely used in traditional medicine”, instead of “P. japonica Houtt., a traditional medicine..”

Line 155:         Please explicate the acronym NASH.

Line 162:         This sentence should be changed to “mechanisms of PC isolated from P. japonica Houtt in adipocytes”

Line 166:         The last part of the sentence should be changed to “dexamethasone (glucocorticoid), and insulin[9]”.

Author Response

Hyeung Jin Jang, Ph.D.

Professor

Department of Biochemistry, College of Korean Medicine

Kyung Hee University, 26, KungHeedae-ro, Dondaemun-gu, Seoul, 02447, Korea

E-mail: hjjang@khu.ac.kr

Apr 20th, 2021

Dear International Journal of Molecular Sciences Editor,

We would like to thank you for the letter dated 6th April, 2021, and we hereby resubmit a revised copy of the manuscript for your consideration. We would also like to express our thanks to all the reviewers for the positive feedback and helpful comments.

In this revised manuscript, we responded carefully to the comments by reviewers point by point as shown in Author’s response. We hope that we addressed well to comments and also wish our revised manuscript would be accepted for publication in International Journal of Molecular Sciences.

Best regards,

Hyeung-Jin Jang

Reviewer 1

The paper is interesting and adds important knowledge regarding the anti-obesogenic effects of natural compounds. However, some points should be addressed before considering it suitable for publication in IJMS.

Point 1: Cell viability assay: It would be more appropriate to assess cell viability for 8 days instead of 48h, since all subsequent PC treatments lasted for 8 days. Furthermore, cell viability was assessed on pre-adipocytes (presumably in medium containing 10% FBS and no MDI), while all subsequent experiments with PC were performed on differentiating adipocytes (medium containing MDI or insulin). Susceptibility to PC could be different on pre-adipocytes and differentiating adipocytes.

(Response) Thanks for your comments. As pointed out by the reviewer, we checked the cytotoxicity of PC treatment for 8 days.  As a result, PC showed no significant toxicity when the 3T3-L1 adipocytes were treated with PC concentration up to 50 μM. So, we changed Figure 1B, and the contents related to Figure 1B as well (line 67-71, 80-87, and 260-268).

Point 2: The statement “PC showed a non-toxic effect up to a concentration of 50 μM” (line 68) is not correct: treatments with all PC concentrations show a statistically significant decrease in the number of viable cells (Figure 1B). According to these results, all tested PC concentrations affect cell viability. Please check statistics or otherwise repeat all experiments with non-toxic concentrations.

(Response) Thanks for your critical comments. We re-tested. As a result of this experiment, it was confirmed that PC showed no toxicity and no significance when the 3T3-L1 adipocytes were treated with PC concentration up to 50 μM for 8 days.

Point 3: Figure 5 should be cited in the discussion when appropriate from line 183 to line 213

(Response) The line number was incorrect because of reflecting the reviewer comments. But as pointed out by the reviewer, we cited Figure 5 in discussion line 218-219.

(line 218-219) “Thus, this study suggests that PC downregulates adipocyte differentiation through the AMPK and MAPK/Akt signaling pathways (Figure 5).”

Point 4: Figure 5 is partially incorrect. According to the results, PC activates AMPK phosphorylation (in the figure is the opposite); in turn, AMPK activation should inhibit the

expression of SREBP1c precursor.

(Response) As pointed out by the reviewer, we changed Figure 5 in manuscript. Thank you for your advice.

Point 5: Lines 18-19: Please add the acronym of sterol regulatory element-binding protein 1 and fatty acid synthase.

(Response) As pointed out by the reviewer, we added the acronym (SREBP 1c and FAS).

Point 6: Line 20-21: Please add the acronym of AMP-activated protein kinase/acetyl-CoA and mitogen-activated protein kinase/protein kinase B.

(Response) As pointed out by the reviewer, we added the acronym (AMPK/ACC and MAPK).

Point 7: Line 35: I believe that “stability” is not the correct term here.

(Response) As pointed out by the reviewer, we changed “safety”, not “stability”. Thank you for your detailed advice.

(line: 35-36) “Therefore, many researchers have studied the anti-obesity effects of natural products with superior safety [3, 4].”

Point 8: Line 40: 3T3-L1 pre-adipocytes are used to study the mechanisms involved in adipogenesis and adipocyte lipid accumulation, not obesity (for this, an in vivo model is required)

(Response) As you mentioned, 3T3-L1 cells have been used to investigate the effects of compounds or nutrients on adipogenesis, to establish the underlying mechanisms of adipogenesis, and to evaluate the potential applications of various compounds and nutrients in the treatment of obesity. We changed the sentence in manuscript.

Point 9: Line 42: Please add “medium” after MDI.

(Response) We added.

Point 10: Line 58: It would be more correct to state “P. japonica Houtt., a perennial herbaceous plant widely used in traditional medicine”, instead of “P. japonica Houtt., a traditional medicine.”

(Response) Thanks for your comments. As pointed out by the reviewer, we changed the sentence in the manuscript.

Point 11: Line 155: Please explicate the acronym NASH.

(Response) As pointed out by the reviewer, we added meaning of the term “NASH” (Non-alcoholic steatohepatitis).

Point 12: Line 162: This sentence should be changed to “mechanisms of PC isolated from P. japonica Houtt in adipocytes”

(Response) We changed the sentence in the manuscript.

Point 13: Line 166: The last part of the sentence should be changed to “dexamethasone (glucocorticoid), and insulin [9]”.

(Response) We changed the sentence in the manuscript.

Reviewer 2 Report

The work by Jee and colleagues entitled “Anti-obesity effect of Polygalin C isolated from Polygala japonica Houtt. Via suppression of the adipogenic and lipogenic factors in 3T3-L1 adipocytes” describes the in vitro effects of Polygalin C, a compound extracted from Polygala japonica Houtt on 3T3-L1 adipocytes. The assessment of the potential biological effects of food components and introduce additional mechanisms by which polyphenols contribute to improved health is highly desirable. The manuscript is well structured and text well written.

However, the authors have disregarded some important basic aspects and hence the manuscript is not acceptable in its present form.

Major points:

  • The authors have not taken into account the metabolization of PC upon ingestion. In reality, the compound present in the extract may not reach the adipose tissue in its parent form (PC) but become metabolized as it crosses the GI tract. The authors need to show (using the available literature) that PC reaches adipocytes in its present form.
  • The authors decided to use PC concentration ranging 0-50mM in their incubation experiments. The authors need to show PC reaches adipocytes and exists in this concentration range (mM).
  • The authors decided for a 48hr incubation period of PC for the cell viability assays (Section 4.4) and cell cultures (Section 4.3). What is the rationale behind this? It is widely described the poor chemical stability exhibited by food and fruit polyphenols under neutral cell culture conditions (please see works by Zhou, Q., Chiang, H., Portocarrero, C., Zhu, Y., Hill, S., Heppert, K., … Kissinger, P. American Chemical Society (ACS) (2003) 20, 83; Li, N., Taylor, L. S., Ferruzzi, M. G., & Mauer, L. J. J Agric Food Chem (2012) 60, 12531; Sang, S., Lee, M. J., Hou, Z., Ho, C. T., & Yang, C. S. J. Agric. Food Chem. (2005) 53(24), 9478; Xiao, J., & Högger, P. J. Agric. Food Chem. (2015) 63, 1547; Zeng, L., Ma, M., Li, C., & Luo, L. Int J Food Propert (2017) 20, 1.) The authors need to show that the observed effects are due to PC and not to PC degradation products. The inclusion of an additional HPLC step at the end of the incubation period to quantify the PC-related products in the cell medium may be helpful. Without this step the authors can hardly claim that the effects observed are due to PC.

Author Response

Dear International Journal of Molecular Sciences Editor,

We would like to thank you for the letter dated 6th April, 2021, and we hereby resubmit a revised copy of the manuscript for your consideration. We would also like to express our thanks to all the reviewers for the positive feedback and helpful comments.

In this revised manuscript, we responded carefully to the comments by reviewers point by point as shown in Author’s response. We hope that we addressed well to comments and also wish our revised manuscript would be accepted for publication in International Journal of Molecular Sciences.

Best regards,

Hyeung-Jin Jang

Reviewer 2

The manuscript is well structured and text well written. However, the authors have disregarded some important basic aspects and hence the manuscript is not acceptable in its present form.

Point 1: The authors have not taken into account the metabolization of PC upon ingestion. In reality, the compound present in the extract may not reach the adipose tissue in its parent form (PC) but become metabolized as it crosses the GI tract. The authors need to show (using the available literature) that PC reaches adipocytes in its present form

(Response) Thanks for your comments. As you mentioned, this study confirmed the potential of PC to treat obesity in vitro, but in fact, when ingested PC, the anti-obesity effect through metabolism cannot be confirmed. We are planning an animal experiment to investigate the efficacy of PC as a follow-up experiment.

Point 2: The authors decided to use PC concentration ranging 0-50μM in their incubation experiments. The authors need to show PC reaches adipocytes and exists in this concentration range (mM).

(Response) As pointed out by the reviewer, we retested the cytotoxicity assessment for the corresponding concentration range in 3T3-L1 adipocytes. As a result, it was confirmed that there was no significant toxicity when PC was treated up to 50 μM.

Point 3: The authors decided for a 48hr incubation period of PC for the cell viability assays (Section 4.4) and cell cultures (Section 4.3). What is the rationale behind this? It is widely described the poor chemical stability exhibited by food and fruit polyphenols under neutral cell culture conditions (please see works by Zhou, Q., Chiang, H., Portocarrero, C., Zhu, Y., Hill, S., Heppert, K., Kissinger, P. American Chemical Society (ACS) (2003) 20, 83; Li, N., Taylor, L. S., Ferruzzi, M. G., & Mauer, L. J. J Agric Food Chem (2012) 60, 12531; Sang, S., Lee, M. J., Hou, Z., Ho, C. T., & Yang, C. S. J. Agric. Food Chem. (2005) 53(24), 9478; Xiao, J., & Högger, P. J. Agric. Food Chem. (2015) 63, 1547; Zeng, L., Ma, M., Li, C., & Luo, L. Int J Food Propert (2017) 20, 1.) The authors need to show that the observed effects are due to PC and not to PC degradation products. The inclusion of an additional HPLC step at the end of the incubation period to quantify the PC-related products in the cell medium may be helpful. Without this step the authors can hardly claim that the effects observed are due to PC.

(Response) Thanks for your critical advice. First, the contents of the cell viability assay were changed by revision. We checked the cytotoxicity of PC treatment for 8 days. As a result, PC showed no toxicity up to 50 μM in 3T3-L1 adipocytes, and all relevant contents were changed.

Second, we checked the reference that you sent, and we found that it is not possible to know whether it is the effect of PC or PC metabolites through this study. We would investigate the content of metabolites in detail in the evaluation of the efficacy of the PC in the animal experiment planning as follow-up experiment.

Reviewer 3 Report

In this present study investigated “Anti-obesity effect of Polygalin C isolated from Polygala japon-2 ica Houtt. Via suppression of the adipogenic and lipogenic fac-3 tors in 3T3-L1 adipocytes”. There are some concerns to be addressed:

  1. 3T3-L1 differentiation procedure is very strange. The culture medium should be replaced with DMEM containing 10 ug/mL insulin for 2 days. However, In this manuscript, the media was replaced by maintenance medium containing 1 μg/mL insulin and PC (0–50 μM) once every 2 days (from Day2 to Day 8).

2.Figure 1D. Oil Red O stain is not clear in this manuscript.

3.Figure 2 B-E detect PPARγ ,C/EBPα, C/EBPβ and C/EBPδ gene expression. However, Figure 1A only detect PPARγ , C/EBPα protein expression. I suggested that detected C/EBPβ and C/EBPδ protein expression.

4.Figure 4, The image is not clear. I did not confirm that srebp1c translocate into nuclear.

  1. Polygalin C can decrease lipid accumulation in adipocyte via blocked lipogenesis.  However, the molecular mechanism is not enough to confirm the conclusion.  Authors should use inhibitor or siRNA to provide more experimental evidence.  Authors also should to detect lipolysis and Fatty acid b-oxidation pathway in 3T3-L1 cells. International Journal of Molecular Sciences is a high-quality journal. Hence, I suggested that should provide more results of animal experiments in obese

Mice.

Round 2

Reviewer 1 Report

The manuscript can now be published in the present form

Reviewer 2 Report

The authors have made slight changes to the revised version however the authors have not included: 

1) any published literature on the metabolization of PC (or similar compounds) that could justify that PC reaches adipose tissue,

2) concentration values of PC that could justify the 0-50uM range tested,

3) studied the chemical stability of PC in the cell medium used to justify that PC does not degrade during the incubation period.

In response, the authors themselves state that "...it is not possible to know whether it is the effect of PC or PC metabolites"